

# Circulating microRNAs demonstrate limited diagnostic potential for diabetic retinopathy in the population of Kazakhstan

Aizhan Magazova[1,2,3], Yeldar Ashirbekov[1], Arman Abaildayev[1], Kantemir Satken[1], Gulzhakhan Utegenova[1,4], Ayaz Belkozhayev[1,5], Altynay Balmukhanova[6], Zaure Dzhumatayeva[7], Ainagul Beissova[8], Iryna Shargorodska[9], Aigul Balmukhanova[10] and Kamalidin Sharipov[1,11]

[1] Aitkhozhin Institute of Molecular Biology and Biochemistry, Almaty, Kazakhstan
[2] Almaty Multidisciplinary Clinical Hospital, Almaty, Kazakhstan
[3] Department of Ophthalmology, Asfendiyarov Kazakh National Medical University, Almaty, Kazakhstan
[4] Department of Biology, South Kazakhstan Pedagogical University named after Ozbekali Zhanibekov, Shymkent, Kazakhstan
[5] Department of Chemical and Biochemical Engineering, Geology and Oil-Gas Business Institute named after K. Turyssov, Satbayev University, Almaty, Kazakhstan
[6] Department of Health Policy and Organization, Al-Farabi Kazakh National University, Almaty, Kazakhstan
[7] Kazakh Scientific Research Institute of Eye Diseases, Almaty, Kazakhstan
[8] Department of Public Health, Al-Farabi Kazakh National University, Almaty, Kazakhstan
[9] Ophthalmology and Optometry department of Postgraduate Education, Bogomolets National Medical University, Kyiv, Ukraine
[10] International Medical School, Caspian University, Almaty, Kazakhstan
[11] Department of Biochemistry, Asfendiyarov Kazakh National Medical University, Almaty, Kazakhstan

Corresponding author
Yeldar Ashirbekov, eldarasher@mail.ru

## ABSTRACT

**Background:** Diabetic retinopathy (DR) is the most common complication of diabetes, leading to blindness. The asymptomatic onset and the existing difficulties in diagnosing warrant the search for biomarkers that can facilitate the early diagnosis of DR. The aim of this study was to evaluate the potential of plasma microRNAs (miRNAs), which have previously been shown to be involved in the pathogenesis of DR and differentially expressed in plasma/serum of patients, as biomarkers for DR in the Kazakhstani population.

**Materials and Methods:** Using quantitative RT-PCR, we compared the levels of ten candidate miRNAs in plasma among three groups: type 2 diabetes mellitus (T2DM) patients with DR (DR patients, $N = 100$), T2DM patients without DR (noDR patients, $N = 98$), and healthy controls ($N = 30$).

**Results:** Level of miR-423-3p was significantly reduced in DR patients compared to noDR patients ($p_{FDR} = 5.4 \times 10^{-3}$). Levels of miR-423-3p and miR-221-3p were significantly reduced in DR patients compared to controls ($p_{FDR} = 5.4 \times 10^{-3}$ and 0.024, respectively), level of miR-23a-3p was significantly reduced in noDR patients compared to controls ($p_{FDR} = 0.047$), levels of miR-221-3p and miR-23a-3p were significantly reduced in T2DM patients (combined group) compared to controls

($p_{FDR}$ = 0.047, and 0.049, respectively). Also, there were several significant differences between groups formed based on clinical-pathological characteristics, but none of these results remained significant after adjustment for multiple comparisons. Correlation analysis revealed weak associations between the levels of miR-423 and miR-221-3p and DR staging ($p_{FDR}$ = $1.3 \times 10^{-3}$ and 0.026, respectively), and fair associations between the levels of miR-29b-3p and miR-328-3p and diabetes duration in noDR patients ($p_{FDR}$ = $8.8 \times 10^{-3}$ and 0.016, respectively). According to receiver operating characteristic (ROC) analysis, only miR-23a-3p can be considered a potential biomarker with moderate informativeness for diagnosing proliferative DR (PDR); however, a larger sample size is needed to verify this finding. Furthermore, the small magnitude of observed changes in miRNA levels between groups significantly complicates classification.

**Conclusions:** Due to the low specificity and small magnitude of deviations from the norm, the studied miRNAs have low potential in the diagnosis of DR.

## INTRODUCTION

The global incidence of type 2 diabetes mellitus (T2DM) is on the rise, with projections indicating a significant increase (*WHO, 2020*; *International Diabetes Federation, 2021*). Diabetic retinopathy (DR) is a microvascular complication of diabetes and is one of the leading causes of blindness in working-age individuals worldwide. DR develops as a result of damage to small blood vessels in the retina due to prolonged hyperglycemia in people with diabetes. DR is characterized by retinal vascular permeability, retinal ischemia, angiogenesis, and inflammation. These pathologies clinically manifest as cotton wool spots, exudates, small tortuous veins, aneurysms, and areas of hemorrhage, which lead to decreased visual acuity, color vision deficiency, and compromised night vision (*Cheung, Mitchell & Wong, 2010*; *Gardner & Sundstrom, 2017*; *WHO, 2020*). Retinal inflammation in DR induces vascular permeability and compromises the integrity of the blood-brain barrier, culminating in the onset of diabetic macular edema (DME), which detrimentally impacts central vision (*Cheung, Mitchell & Wong, 2010*; *Klaassen, Van Noorden & Schlingemann, 2013*; *Browning, Stewart & Lee, 2018*). Another complication arises from retinal ischemia, prompting the formation of aberrant new blood vessels within the central posterior segment—a condition known as proliferative DR (PDR). These vessels, prone to attachment to the vitreous surface, are susceptible to rupture, potentially leading to retinal detachment and subsequent vision impairment (*Cheung, Mitchell & Wong, 2010*; *Chaudhary, Zaveri & Becker, 2021*).

Traditional risk factors associated with DR development encompass the duration of diabetes, hyperglycemia, hypertension, and dyslipidemia. Vigilant management of these risk factors often facilitates the halting or decelerating of disease progression in its early

stages. However, despite favorable prognoses, some patients still develop DR, while others with poor prognoses remain unaffected, indicating a genetic component to the risk of DR (*Keenan et al., 2007*; *Sivaprasad et al., 2012*; *Singh et al., 2021*). Current treatment modalities for DR, such as laser therapy and intraocular drug injections, can mitigate further visual deterioration, but they have many adverse effects (*Simó-Servat, Simó & Hernández, 2016*). It has also been shown a rapid progression of the disease to proliferative DR (PDR) (*Yun et al., 2016*). Thus, the detection of early DR is a key point for preserving vision. However, the asymptomatic onset of the disease and the variability of clinical signs present diagnostic complexities, underscoring the necessity for healthcare practitioners to possess high levels of expertise (*WHO Regional Office for Europe, 2020*; *Vujosevic et al., 2020*). This serves as the foundation for the development of new methods for early diagnosis and prognosis of DR, including those based on biomarkers. Several different classes of molecules involved in various processes related to diabetes and its complications, including microRNAs (miRNAs), are being considered as biomarkers for DR (*Jenkins et al., 2015*; *Simó-Servat, Simó & Hernández, 2016*; *Ting et al., 2016*).

miRNAs represent a class of short non-coding RNA molecules that play a key role in the regulation of gene expression, thus exerting an important function in a variety of biological processes, including embryogenesis, cell differentiation, metabolism, and apoptosis (*Ha & Kim, 2014*). Abnormalities in miRNA regulation may be associated with various diseases, including cancer (*Peng & Croce, 2016*), cardiovascular disease (*Colpaert & Calore, 2019*), autoimmune diseases (*Long et al., 2018*), neurological disorders (*Swarbrick et al., 2019*), diabetes and its complications (*Gong et al., 2017*; *Deng & Guo, 2019*; *He et al., 2021*; *Zhao et al., 2022*). Recorded alterations in the miRNA profile may indicate the presence of a specific disease, therefore miRNAs are being actively studied as potential biomarkers of pathological conditions. miRNAs enter biofluids, being released from cells as part of exosomes (*Sun et al., 2018*), and they are quite stable there (*Grasedieck et al., 2012*; *Glinge et al., 2017*). This makes it possible to determine the level of such a biomarker in a minimally invasive way, which is quite critical in the case of DR.

Previous studies have identified the involvement of certain miRNAs in the pathogenesis of DR (*Gong et al., 2017*; *Kamalden et al., 2017*; *Maisto et al., 2020*; *Kaur et al., 2022*), with many miRNAs being recommended as diagnostic or prognostic markers for DR, including in plasma (*Liu et al., 2018*; *Sangalli et al., 2020*; *Shi et al., 2021*; *Kaur et al., 2022*; *Ko et al., 2022*; *Rezazadeh-Gavgani et al., 2023*). Additionally, evidence suggests the existence of ethnic disparities in the prevalence and severity of DR (*Wong et al., 2006*; *Raymond et al., 2009*; *Sivaprasad et al., 2012*; *Singh et al., 2021*; *Chia et al., 2023*), which extends to the applicability of biomarkers (*Fickweiler et al., 2023*). Therefore, miRNA markers need to be validated for specific ethnic groups.

The aim of our study was to assess the potential of circulating miRNAs as a biomarker of DR in the population of Kazakhstan. To do this, we compared the plasma profiles of certain miRNAs, which, according to the literature data, are involved in the pathogenesis of DR and are dysregulated in plasma/serum of patients, in three groups—T2DM patients with DR, T2DM patients without DR, and healthy controls.

## MATERIALS AND METHODS

### Subjects

Venous blood sampling for the study was conducted at four medical institutions. Blood samples from 100 T2DM patients with clinically confirmed DR (DR patients) and 98 T2DM patients without DR (noDR patients) were collected between 2020–2021 at Almaty Multidisciplinary Clinical Hospital, City Center of Diabetes and Kazakh Research Institute of Eye Diseases of Ministry of Healthcare of the Republic of Kazakhstan. All recruited patients did not have COVID-19, as confirmed by polymerase chain reaction (PCR) tests. Biomaterial from 30 healthy controls was collected in 2022 from Almaty Multidisciplinary Clinical Hospital.

The study was approved by the local ethics committee of Aitkhozhin Institute of Molecular Biology and Biochemistry (approval number 4.60/01-03). Patients with a verified diagnosis of type 2 diabetes mellitus who provided written consent were enrolled in the study. The exclusion criteria were as follows: terminal stage of the disease, type 1 diabetes mellitus, pregnancy and lactation, childhood, cancer, severe cardiovascular, respiratory, renal, infectious, and mental diseases. All the patients underwent visometry, tonometry, biomicroscopy, wide-pupil fundus examination using an aspheric lens, and optical coherence tomography. Patients were divided into four groups according to the classification of Kohner E. and Porta M.: no retinopathy, non-proliferative DR (NPDR), pre-proliferative DR (PPDR), and proliferative DR (PDR).

### Selection of miRNAs

Based on the literature data, nine candidate miRNAs were selected: miR-150-5p (*Mazzeo et al., 2018*; *Yu et al., 2020a*; *Wang, Wu & Wang, 2021*; *Ko et al., 2022*), miR-21-5p (*Qing et al., 2014*; *Chen et al., 2017*; *Jiang et al., 2017*; *Lou et al., 2019*; *Grieco et al., 2022*; *Rezazadeh-Gavgani et al., 2023*; *Wróblewski et al., 2023*), miR-24-3p (*Mazzeo et al., 2018*; *Guo et al., 2021b*; *Grieco et al., 2022*), miR-29b-3p (*Zhang et al., 2017*; *Dantas da Costa e Silva et al., 2019*; *Zeng et al., 2020*), miR-423-3p (*Meerson et al., 2019*; *Blum et al., 2019*), miR-15a-5p (*Kamalden et al., 2017*; *Gong et al., 2019*; *Liu et al., 2019*; *Sangalli et al., 2020*; *Mastropasqua et al., 2021*; *Grieco et al., 2022*), miR-221-3p (*Liu et al., 2018*; *Wang et al., 2020a*; *Zhao & Pan, 2023*), miR-23a-3p (*Yang et al., 2014*; *Mastropasqua et al., 2021*; *Sun, Liu & Zuo, 2021*; *Santovito et al., 2021*), miR-26b-5p (*Shi et al., 2021*; *Zhang et al., 2022*). As endogenous controls, miR-328-3p (*Prado et al., 2019*) and miR-191-5p (*Hu et al., 2012*; *Zheng et al., 2013*; *Donati, Ciuffi & Brandi, 2019*; *Wang et al., 2020b*) were selected, and ath-miR-159a (*Marabita et al., 2016*) was chosen as an exogenous spike-in control.

### Plasma preparation and RNA extraction

Plasma was obtained within 2 h after blood collection, provided it was stored at 4 °C. The blood sample was centrifuged at 1,000 g for 15 min at 4 °C, and the upper layer of plasma was collected. The obtained plasma was then subjected to a second centrifugation at 2,500 g for 15 min at 4 °C, after which the upper fraction was collected, aliquoted into 200 μl portions, and stored at −70 °C until further analysis.
Total RNA was extracted using the MagMAX mirVana Total RNA Isolation Kit (A27828; Applied Biosystems, Waltham, MA, USA) according to the manufacturer's protocol.

## Reverse transcription and quantitative polymerase chain reaction (qRT-PCR)

Complementary DNA (cDNA) was obtained using the TaqMan Advanced miRNA cDNA Synthesis Kit (A28007; Applied Biosystems, Waltham, MA, USA) according to the manufacturer's protocol. The quantitative PCR was performed in a volume of 12 μl using the TaqMan Advanced miRNA Assays kit (A25576; Applied Biosystems, Waltham, MA, USA) and the TaqMan Fast Advanced Master Mix reagent (4444964; Applied Biosystems, Waltham, MA, USA) according to the manufacturer's protocol on the StepOnePlus Real-Time PCR System (4376600; Applied Biosystems, Waltham, MA, USA).

Primary processing was performed using StepOnePlus 2.2.2 and ExpressionSuite v1.3 programs. Relative quantification is carried out using the comparative threshold cycle (ΔΔCt) method with modifications as described by *Königshoff et al. (2009)*. Relative transcript abundance is expressed in ΔCt values ($\Delta Ct = Ct_{reference} - Ct_{target}$). Endogenous miR-191-5p was used as a reference. The suitability of controls was assessed using the programs NormFinder v.20 (*Andersen, Jensen & Ørntoft, 2004*) and GeNorm v.3 (*Vandesompele et al., 2002*). ΔΔCt value ($\Delta\Delta Ct = mean\ \Delta Ct_{case} - mean\ \Delta Ct_{control}$) was considered as $\log_2$ fold change.

## Statistical analysis

Most statistical calculations were performed in the Jamovi v2.2.5 program (*The Jamovi Project, 2024*, https://www.jamovi.org). To compare the characteristics of the studied groups, we used the Mann-Whitney $U$ test for quantitative data and the Pearson's goodness-of-fit test ($\chi^2$-test) for nominal data. The statistical significance of differences in miRNA levels between the groups was assessed using the Mann-Whitney $U$ test. For a comparative visualization of miRNA levels, box plots were constructed using a web tool BoxPlotR (BoxPlotR: a web-tool for generation of box plots. http://shiny.chemgrid.org/boxplotr (accessed on 26 March 2025)). Spearman's rank correlation was used to examine the relationship between variables. Interpretation of correlation coefficients as the strength of the linear relationship was performed as described by *Chan (2003)*. Multiple linear regression was used to evaluate the influence of variables (age, gender, body mass index (BMI), ethnicity, blood glucose levels, disease status, disease duration, and medications taken) on the levels of miRNAs that exhibited differential expression between the groups. $P$-values <0.05 were considered statistically significant. For multiple comparisons, the online calculator of false discovery rate (FDR) correction was used to adjust the $p$-values (Online calculator of FDR correction for multiple comparisons. https://www.sdmproject.com/utilities/?show=FDR (accessed on 26 March 2025)). The characteristics of the markers were assessed based on the results of receiver operating characteristic (ROC) analysis, which was calculated using the online resources easyROC (*Goksuluk et al., 2016*)

and Jamovi. Youden's index method was used to calculate the optimal cut-off point. Evaluation of classifiers by interpretation of the area under the ROC curve (AUC) was performed as described by *Muller et al. (2005)*.

## RESULTS

### Characteristics of the compared groups

The characteristics of the compared groups are presented in Table 1. Healthy controls did not significantly differ in age, gender distribution, and BMI from both groups of diabetic patients. Among the controls, there were significantly more smokers and alcohol drinkers compared to DR patients ($p_{FDR} = 6.5 \times 10^{-4}$ and 0.048, respectively). The main reason for these differences is likely the necessity for patients to abandon harmful habits.

The two groups of diabetic patients did not differ from each other in age, gender distribution, number of alcohol drinkers, number of patients with a family history of diabetes, age of diabetes onset, BMI, and blood glucose levels. There were significantly fewer smokers among DR patients compared to noDR patients ($p_{FDR} = 0.048$). The proportion of medications taken significantly differed between the two groups of diabetic patients ($p_{FDR} = 6.5 \times 10^{-4}$). Moreover, at the time of sample collection, DR patients had a significantly longer duration of diabetes compared to noDR patients ($p_{FDR} = 9.0 \times 10^{-5}$).

### Selecting a normalizer

Using the qRT-PCR method, we determined levels (Ct value) of studied miRNAs in plasma of all individuals from three groups (File S1). According to the literature, two endogenous controls (miR-328-3p and miR-191-5p) and one exogenous control (ath-miR-159a) were selected for normalizing quantitative data. We assessed the suitability of the controls using known programs, NormFinder v.20 and GeNorm v.3.

NormFinder allows for calculating variation between two compared groups. Since we had three studied groups, by combining them, we obtained three comparisons: DR patients *vs.* noDR patients + controls, DR patients + noDR patients *vs.* controls, DR patients + controls *vs.* noDR patients. In addition, we performed calculations without considering comparison groups. In all four data loading scenarios, the best stability values were obtained for miR-191-5p, slightly worse results for miR-328-3p, and much worse results for the exogenous ath-miR-159a. GeNorm showed similar results (File S2).

The developers of GeNorm recommend using at least three genes for normalizing quantitative data. However, considering the expression stability data, we excluded ath-miR-159a (due to its insufficient stability), and considering the transcript abundance, we excluded miR-328-3p (due to its insufficient abundance) from the list of controls. Thus, only miR-191-5p was used as a normalizer, while miR-328-3p was included to the list of candidate miRNAs.

### Comparative miRNA levels between groups

Comparative quantitative statistics between the three groups are presented in Table 2. Comparison of the levels of studied miRNAs between groups is selectively presented in Fig. 1. First, we compared the two groups of diabetic patients. The level of miR-423-3p

**Table 1 Demographic and clinical characteristics of the study groups.**

| Characteristics | DR patients (A) | noDR patients (B) | Controls (C) | A vs. B p-value | A vs. C p-value | B vs. C p-value |
|---|---|---|---|---|---|---|
| Total (No.) | 100 | 98 | 30 | | | |
| Of them: | | | | | | |
| Female sex (%) | 63.0 | 57.1 | 50.0 | 0.400 | 0.202 | 0.491 |
| Kazakhs (%) | 65.0 | 68.4 | 93.3 | – | – | – |
| Uighurs (%) | 15.0 | 14.3 | 0 | – | – | – |
| Russians (%) | 14.0 | 9.2 | 3.3 | – | – | – |
| Other ethnic groups[§] (%) | 6.0 | 8.2 | 3.3 | – | – | – |
| Smokers (%) | 8.0 | 20.4 | 36.7 | 0.012* | $9.7 \times 10^{-5}$* | 0.069 |
| Drinkers (on holidays) (%) | 13.0 | 14.3 | 33.3 | 0.792 | 0.011* | 0.019 |
| With a family history of DM (%) | 45.0 | 43.9 | – | 0.874 | – | – |
| Medications: insulin/medformin/other/no (%) | 78.0/11.0/10.0/1.0 | 49.0/33.7/11.2/6.1 | – | $7.4 \times 10^{-5}$* | – | – |
| NPDR[1] (%) | 38.0 | – | – | – | – | – |
| PPDR[2] (%) | 40.0 | – | – | – | – | – |
| PDR[3] (%) | 22.0 | – | – | – | – | – |
| DME[4] (%) | 20.0 | – | – | – | – | – |
| Age (years) | 60.31 ± 10.48 | 59.48 ± 9.62 | 59.73 ± 6.98 | 0.673 | 0.742 | 0.842 |
| Age of diabetes onset (yrs) | 47.14 ± 12.24 | 50.01 ± 10.85 | – | 0.032 | – | – |
| Diabetes duration (yrs) | 12.64 ± 6.60 | 8.48 ± 7.21 | – | $4.5 \times 10^{-6}$* | – | – |
| Body mass index | 28.22 ± 4.35 | 28.17 ± 5.60 | 27.71 ± 4.45 | 0.750 | 0.689 | 0.914 |
| Blood glucose (fasting) (mmol/L) | 9.80 ± 3.66 | 9.77 ± 3.34 | – | 0.755 | – | – |

Notes:
[§] Koreans, Kurds, Azerbaijanis, Uzbeks, Karakalpaks, Tatars, Udmurts, Turks, Chinese.
[1] Non-proliferative DR.
[2] Pre-proliferative DR.
[3] Proliferative DR.
[4] Diabetic macular edema.
* p-value remains significant after FDR correction for multiple comparisons.

was significantly decreased in the plasma of DR patients compared to noDR patients ($p = 2.1 \times 10^{-4}$), although the fold change was very minimal—only 1.2-fold differences. This difference remained significant after adjustment for multiple comparisons ($p_{FDR} = 5.4 \times 10^{-3}$). The levels of the other ten miRNAs did not significantly differ between these two groups of patients. Nevertheless, it is worth noting that two miRNAs were close to being recognized as altered in DR: the level of miR-29b-3p was increased and the level of miR-221-3p was decreased in DR patients compared to noDR patients; the significance in both comparisons did not reach the accepted threshold, but was below 0.1 ($p = 0.075$ and 0.098, respectively).

Next, we compared two groups of diabetic patients with healthy controls. It was found that the levels of miR-423-3p, miR-221-3p, and miR-23a-3p were significantly decreased in DR patients compared to controls ($p = 2.7 \times 10^{-4}$, $1.8 \times 10^{-3}$, and 0.022, respectively); the first two differences remained significant after adjustment for multiple comparisons ($p_{FDR} = 5.4 \times 10^{-3}$, 0.024 and 0.126, respectively). The largest fold change, almost reaching 1.5-fold differences, was observed for miR-221-3p.

**Table 2  Cycle threshold values (Ct) and comparative statistics of studied miRNAs between DR patients, noDR patients, and controls.**

| miRNA | Ct mean ± SD (ΔCt mean ± SE) | | | DR patients vs. noDR patients | | DR patients vs. controls | | noDR patients vs. controls | | T2DM patients vs. controls | |
|---|---|---|---|---|---|---|---|---|---|---|---|
| | DR patients | noDR patients | Controls | ΔΔCt (95% CI) | p-value | ΔΔCt (95% CI) | p-value | ΔΔCt (95% CI) | p-value | ΔΔCt (95% CI) | p-value |
| miR-150-5p | 21.91 ± 1.23 (1.28 ± 0.09) | 22.47 ± 1.47 (1.35 ± 0.10) | 22.62 ± 1.70 (1.62 ± 0.14) | −0.09 [−0.34 to 0.16] | 0.512 | −0.30 [−0.64 to 0.06] | 0.113 | −0.20 [−0.58 to 0.12] | 0.239 | −0.25 [−0.58 to 0.06] | 0.140 |
| miR-21-5p | 21.14 ± 1.14 (2.04 ± 0.07) | 21.95 ± 1.38 (1.86 ± 0.07) | 22.27 ± 1.72 (1.98 ± 0.10) | 0.16 [−0.04 to 0.36] | 0.114 | 0.03 [−0.21 to 0.30] | 0.789 | −0.13 [−0.39 to 0.13] | 0.338 | −0.04 [−0.28 to 0.20] | 0.718 |
| miR-24-3p | 22.61 ± 1.12 (0.57 ± 0.06) | 23.31 ± 1.40 (0.51 ± 0.06) | 23.49 ± 1.68 (0.75 ± 0.09) | 0.06 [−0.10 to 0.22] | 0.480 | −0.16 [−0.38 to 0.05] | 0.147 | −0.22 [−0.45 to 0.00] | 0.049 | −0.19 [−0.40 to 0.01] | 0.069 |
| miR-29b-3p | 23.88 ± 1.37 (−0.70 ± 0.07) | 24.65 ± 1.60 (−0.84 ± 0.07) | 25.08 ± 1.94 (−0.84 ± 0.12) | 0.19 [−0.02 to 0.38] | 0.075 | 0.18 [−0.09 to 0.41] | 0.169 | −0.02 [−0.29 to 0.29] | 0.859 | 0.09 [−0.18 to 0.34] | 0.519 |
| miR-328-3p | 27.82 ± 1.45 (−4.63 ± 0.06) | 28.43 ± 1.50 (−4.61 ± 0.07) | 28.79 ± 1.89 (−4.55 ± 0.14) | −0.03 [−0.23 to 0.16] | 0.708 | −0.07 [−0.39 to 0.23] | 0.685 | −0.05 [−0.38 to 0.27] | 0.802 | −0.06 [−0.37 to 0.24] | 0.725 |
| miR-423-3p | 25.98 ± 1.30 (−2.80 ± 0.05) | 26.33 ± 1.40 (−2.51 ± 0.06) | 26.71 ± 1.51 (−2.47 ± 0.08) | −0.28 [−0.43 to −0.14] | $2.1 \times 10^{-4}$* | −0.34 [−0.50 to −0.16] | $2.7 \times 10^{-4}$* | −0.04 [−0.25 to 0.16] | 0.607 | −0.21 [−0.38 to −0.02] | 0.026 |
| miR-15a-5p | 20.37 ± 1.50 (2.88 ± 0.08) | 21.01 ± 1.74 (2.81 ± 0.07) | 21.37 ± 1.51 (3.02 ± 0.10) | 0.05 [−0.15 to 0.25] | 0.591 | −0.14 [−0.41 to 0.12] | 0.291 | −0.20 [−0.48 to 0.08] | 0.171 | −0.17 [−0.43 to 0.09] | 0.196 |
| miR-221-3p | 23.02 ± 0.92 (0.22 ± 0.08) | 23.40 ± 1.10 (0.42 ± 0.09) | 23.65 ± 1.16 (0.75 ± 0.13) | −0.21 [−0.44 to 0.04] | 0.098 | −0.55 [−0.85 to −0.20] | $1.8 \times 10^{-3}$* | −0.32 [−0.64 to −0.01] | 0.037 | −0.43 [−0.73 to −0.12] | $5.7 \times 10^{-3}$* |
| miR-23a-3p | 24.15 ± 1.08 (−0.90 ± 0.07) | 24.83 ± 1.32 (−1.01 ± 0.08) | 24.96 ± 1.33 (−0.57 ± 0.11) | 0.10 [−0.10 to 0.30] | 0.330 | −0.31 [−0.56 to −0.04] | 0.022 | −0.40 [−0.69 to −0.12] | $5.9 \times 10^{-3}$* | −0.36 [−0.61 to −0.10] | $7.4 \times 10^{-3}$* |
| miR-26b-5p | 21.97 ± 1.23 (1.27 ± 0.05) | 22.60 ± 1.52 (1.22 ± 0.04) | 23.06 ± 1.47 (1.34 ± 0.06) | 0.06 [−0.07 to 0.18] | 0.324 | −0.02 [−0.19 to 0.13] | 0.847 | −0.09 [−0.24 to 0.06] | 0.264 | −0.05 [−0.21 to 0.09] | 0.484 |
| miR-191-5p | 23.18 ± 1.44 (−) | 23.82 ± 1.65 (−) | 24.24 ± 1.70 (−) | – | – | – | – | – | – | – | – |
| ath-miR-159a | 15.67 ± 2.89 (−) | 12.75 ± 2.84 (−) | 14.04 ± 3.40 (−) | – | – | – | – | – | – | – | – |

**Note:**
* p-Value remains significant after FDR correction for multiple comparisons.

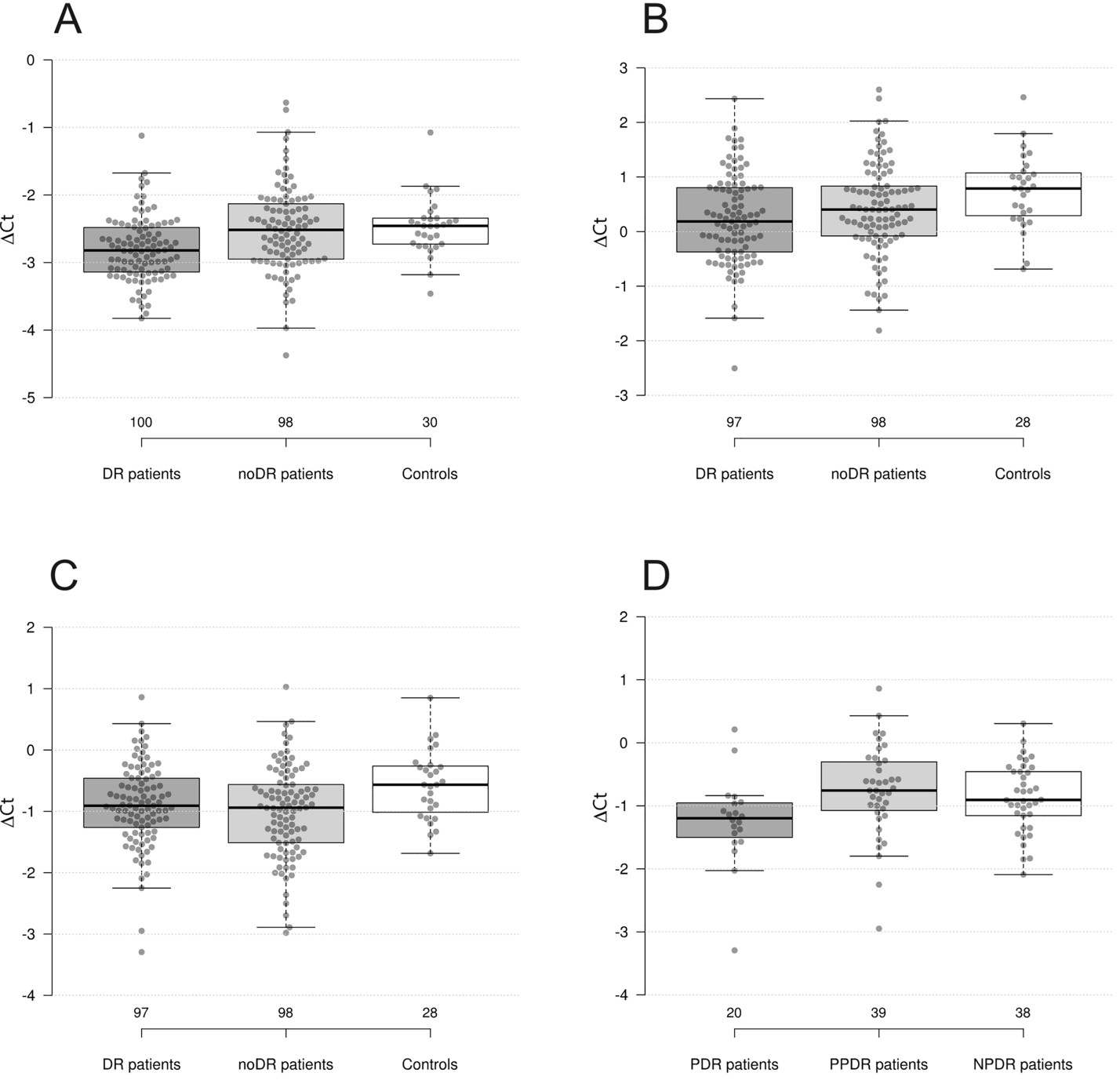

**Figure 1 Differences in levels of miRNAs between groups.** (A) miR-423-3p in DR patients, noDR patients, and controls; (B) miR-221-3p in DR patients, noDR patients, and controls; (C) miR-23a-3p in DR patients, noDR patients, and controls; (D) miR-23a-3p between PDR, PPDR, and NPDR patients.

In the second comparison, the levels of miR-23a-3p, miR-221-3p, and miR-24-3p were significantly decreased in noDR patients compared to controls ($p = 5.9 \times 10^{-3}$, 0.037, and 0.049, respectively); only one difference out of three remained significant after adjustment

Table 3 *p* -values for comparisons between groups with different clinicopathological characteristics.

| Compared groups | miRNA | | | | | | | | | |
|---|---|---|---|---|---|---|---|---|---|---|
| | miR-150-5p | miR-21-5p | miR-24-3p | miR-29b-3p | miR-328-3p | miR-423-3p | miR-15a-5p | miR-221-3p | miR-23a-3p | miR-26b-5p |
| NPDR patients *vs.* noDR patients | 0.956 | 0.023 | 0.411 | 0.058 | 0.913 | 0.016 | 0.336 | 0.584 | 0.343 | 0.214 |
| PPDR patients *vs.* NPDR patients | 0.404 | 0.400 | 0.673 | 0.406 | 0.595 | 0.901 | 0.638 | 0.796 | 0.397 | 0.951 |
| PDR patients *vs.* PPDR patients | 0.901 | 0.140 | 0.069 | 0.948 | 0.153 | 0.309 | 0.584 | 0.131 | $4.1 \times 10^{-3}$ | 0.476 |
| PDR patients *vs.* noPDR patients | 0.930 | 0.040 | 0.063 | 0.730 | 0.242 | 0.251 | 0.399 | 0.057 | $5.1 \times 10^{-3}$ | 0.347 |
| DME patients *vs.* noDME patients | 0.976 | 0.813 | 0.544 | 0.779 | 0.396 | 0.184 | 0.510 | 0.902 | 0.380 | 0.117 |
| DR patients *vs.* noDR patients with >10 yrs diabetes | 0.149 | 0.719 | 0.687 | 0.376 | 0.032 | 0.012 | 0.185 | 0.208 | 0.264 | 0.693 |
| DR patients, diabetes duration: >10 *vs.* ≤10 yrs | 0.393 | 0.433 | 0.635 | 0.095 | 0.980 | 0.536 | 0.067 | 0.997 | 0.710 | 0.811 |
| noDR patients, diabetes duration: >10 *vs.* ≤10 yrs | 0.197 | 0.036 | 0.867 | 0.010 | 0.011 | 0.745 | 0.024 | 0.963 | 0.641 | 0.081 |
| DR patients, BMI: >30 *vs.* ≤30 | 0.856 | 0.676 | 0.839 | 0.731 | 0.839 | 0.915 | 0.122 | 0.953 | 0.816 | 0.049 |
| noDR patients, BMI: >30 *vs.* ≤30 | 0.269 | 0.669 | 0.646 | 0.592 | 0.500 | 0.666 | 0.078 | 0.994 | 0.826 | 0.139 |
| DR patients, family history of diabetes: Yes *vs.* No | 0.542 | 0.458 | 0.638 | 0.923 | 0.906 | 0.923 | 0.493 | 0.201 | 0.436 | 0.514 |
| noDR patients, family history of diabetes: Yes *vs.* No | 0.611 | 0.210 | 0.359 | 0.461 | 0.173 | 0.889 | 0.892 | 0.631 | 0.621 | 0.392 |
| Controls, family history of diabetes: Yes *vs.* No | 0.493 | 0.189 | 0.446 | 0.704 | 0.085 | 0.164 | 0.940 | 0.140 | 0.140 | 0.595 |

for multiple comparisons ($p_{FDR}$ = 0.047, 0.164 and 0.196, respectively). The largest fold change for miR-23a-3p slightly exceeded 1.3-fold differences.

We also compared the combined sample of diabetes patients (with and without DR) to healthy controls. As a result, three out of four dysregulated miRNAs retained their status: levels of miR-221-3p, miR-23a-3p, and miR-423-3p were significantly decreased in diabetes patients compared to healthy controls ($p = 5.7 \times 10^{-3}$, $7.4 \times 10^{-3}$, and 0.026, respectively); two differences remained significant after adjustment for multiple comparisons ($p_{FDR}$ = 0.047, 0.049 and 0.130, respectively). The level of miR-24-3p was also decreased, although not significantly, but close to the accepted threshold ($p = 0.069$).

## Associations with clinicopathological parameters

Results of comparisons between groups formed based on clinical-pathological characteristics are presented in Table 3. Although we found several significant differences, none of these results remained significant after adjustment for multiple comparisons. Nevertheless, we provide descriptions of the results here.

Our main focus was on developing a marker for early diagnosis of DR, so we initially compared patients with early-stage DR (NPDR) to noDR patients. We found that the level of miR-423-3p was significantly decreased, while the level of miR-21-5p was significantly increased in NPDR patients compared to noDR patients ($p = 0.16$ and 0.23, respectively). It

is also noteworthy that miR-29b-3p was elevated in the plasma of NPDR patients, although the *p*-value was slightly above the significance threshold ($p = 0.058$).

Next, we continued the comparisons between the groups according to the degree of DR progression. None of the miRNAs studied showed differential expression between the NPDR and PPDR patient groups. miR-23a-3p was significantly decreased in PDR patients compared to PPDR patients ($p = 4.1 \times 10^{-3}$). miR-23a-3p and miR-21-5p were significantly decreased in PDR patients compared to patients with earlier stages of DR (NPDR+PPDR) ($p = 5.1 \times 10^{-3}$ and 0.040, respectively). We did not find differences in miRNAs' levels between groups based on the presence/absence of DME.

Further we conducted group comparisons based on the duration of diabetes. We observed significantly elevated levels of miR-29b-3p, miR-328-3p, miR-15a-5p, and miR-21-5p in noDR patients with a diabetes duration of more than 10 years compared to noDR patients with a diabetes duration of less than 10 years ($p = 0.010, 0.011, 0.024$, and 0.036, respectively). However, no such differences were observed among DR patients. To test the hypothesis of why some diabetic patients do not develop DR despite long disease duration, we compared noDR patients with disease duration of over 10 years to DR patients. We found that miR-423-3p and miR-328-3p were significantly reduced in DR patients ($p = 0.012$ and 0.032, respectively).

None of the studied miRNAs exhibited differential expression depending on the presence or absence of a family history of diabetes or alcohol consumption status in the three groups. Compared to non-smokers, miR-328-3p was decreased in smokers among noDR patients ($p = 0.044$), miR-328-3p and miR-21-5p were decreased in smokers among healthy controls ($p = 0.023$ and 0.014, respectively).

Finally, differences in miRNA levels were found depending on sex. miR-150-5p was significantly decreased in men compared to women among DR patients ($p = 0.036$), miR-328-3p was significantly decreased in men compared to women among healthy controls ($p = 0.029$).

### Correlation analysis

As a result of the correlation analysis, 31 correlations were identified (Table 4). None of these correlations were "strong" or "good" (rho > |0.5|), five correlations were "fair" (rho between |0.3| and |0.5|), while the rest were "poor" (rho < |0.3|), according to the grading provided by *Chan (2003)*. Only six correlations remained significant after adjustment for multiple comparisons. We describe only these. The levels of miR-423 and miR-221-3p negatively "poorly" correlated with DR staging ($p_{FDR} = 1.3 \times 10^{-3}$ and 0.026, respectively), and the correlation between miR-423-3p and DR staging persisted with the exclusion of the healthy stage ($p_{FDR} = 8.8 \times 10^{-3}$). Levels of miR-29b-3p and miR-328-3p positively "fairly" correlated with diabetes duration in noDR patients ($p_{FDR} = 8.8 \times 10^{-3}$ and 0.016, respectively); the correlation of miR-29b-3p with disease duration persisted in the combined sample of diabetic patients ($p_{FDR} = 3.4 \times 10^{-3}$), although the strength of the association decreased.

**Table 4 Detected correlations between miRNA levels and clinic-pathological parameters.**

| Group | Parameter 1 | Parameter 2 | Spearman rho | p-value |
|---|---|---|---|---|
| DR patients | miR-423-3p | DR stage[1] | −0.294 | $6.3 \times 10^{-6}$* |
| | miR-423-3p | DR stage[2] | −0.269 | $1.3 \times 10^{-4}$* |
| | miR-221-3p | DR stage[1] | −0.225 | $7.3 \times 10^{-4}$* |
| | miR-423-3p | Age | 0.303 | 0.002 |
| | miR-221-3p | Age of diabetes onset | 0.245 | 0.016 |
| | miR-221-3p | DR stage[2] | −0.152 | 0.033 |
| | miR-26b-5p | BMI | −0.217 | 0.033 |
| | miR-29b-3p | HbA1c level[3] | −0.358 | 0.035 |
| | miR-15a-5p | BMI | −0.210 | 0.039 |
| noDR patients | miR-29b-3p | Diabetes duration | 0.371 | $1.7 \times 10^{-4}$* |
| | miR-328-3p | Diabetes duration | 0.352 | $3.8 \times 10^{-4}$* |
| | miR-21-5p | Diabetes duration | 0.298 | 0.003 |
| | miR-15a-5p | HbA1c level[4] | −0.386 | 0.007 |
| | miR-150-5p | Diabetes duration | 0.258 | 0.010 |
| | miR-21-5p | Age of diabetes onset | −0.251 | 0.013 |
| | miR-26b-5p | BMI | 0.245 | 0.015 |
| | miR-150-5p | BMI | 0.234 | 0.020 |
| | miR-150-5p | Age of diabetes onset | −0.228 | 0.024 |
| | miR-423-3p | Age | 0.225 | 0.026 |
| | miR-15a-5p | BMI | 0.202 | 0.046 |
| all T2DM patients | miR-29b-3p | Diabetes duration | 0.291 | $3.2 \times 10^{-5}$* |
| | miR-423-3p | Age of diabetes onset | 0.209 | 0.003 |
| | miR-21-5p | Diabetes duration | 0.200 | 0.005 |
| | miR-29b-3p | Age of diabetes onset | −0.190 | 0.007 |
| | miR-15a-5p | HbA1c level[5] | −0.289 | 0.009 |
| | miR-328-3p | Diabetes duration | 0.177 | 0.013 |
| | miR-15a-5p | Diabetes duration | 0.175 | 0.014 |
| | miR-29b-3p | HbA1c level[5] | −0.264 | 0.016 |
| | miR-26b-5p | HbA1c level[5] | −0.232 | 0.038 |
| | miR-221-3p | Age of diabetes onset | 0.149 | 0.038 |
| | miR-15a-5p | Age of diabetes onset | −0.142 | 0.048 |

Notes:
[1] DR stage ranking: Healthy controls, noDR patients, NPDR patients, PPDR patients, PDR patients.
[2] DR stage ranking: noDR patients, NPDR patients, PPDR patients, PDR patients.
[3] Based on available data for 35 DR patients.
[4] Based on available data for 48 noDR patients.
[5] Based on available data for 83 T2DM patients.
* p-value remains significant after FDR correction for multiple comparisons.

## Multiple regression analysis

To evaluate the influence of clinical and demographic factors on the levels of miRNAs that exhibited differential expression between the groups, we conducted multiple linear regression (Table 5). In addition to disease status, seven significant variables were selected: quantitative variables included age, BMI, duration of diabetes, and blood glucose levels,
**Table 5 Results of multiple regression analysis assessing the influence of clinical and demographic factors on miRNA levels.**

| Compared groups | miRNA | Standardized regression coefficient for variable (p-value) | | | | | | | | Model fit, $R^2$ |
|---|---|---|---|---|---|---|---|---|---|---|
| | | Age | BMI | Diabetes duration | Blood glucose | Sex[1] | Ethnicity[2] | Disease status[3] | Medications[4] | |
| DR patients vs. noDR patients | miR-423-3p | 0.224 (0.0024*) | −0.028 (0.689) | 0.064 (0.390) | 0.129 (0.071) | −0.002 (0.989) | −0.181 (0.227) | −0.633 (2.4 × 10⁻⁵ *) | 0.213 (0.182) | 0.152 |
| DR patients vs. controls | miR-221-3p | 0.122 (0.157) | −0.005 (0.952) | – | – | −0.120 (0.500) | −0.490 (0.011) | −1.023 (2.0 × 10⁻⁵ *) | – | 0.189 |
| | miR-423-3p | 0.204 (0.020) | −0.075 (0.388) | – | – | 0.048 (0.787) | −0.361 (0.064) | −0.762 (0.0014*) | – | 0.156 |
| | miR-23a-3p | 0.015 (0.869) | 0.060 (0.500) | – | – | 0.034 (0.853) | −0.580 (0.0041*) | −0.714 (0.0035*) | – | 0.119 |
| noDR patients vs. controls | miR-221-3p | 0.111 (0.223) | 0.014 (0.879) | – | – | −0.008 (0.966) | −0.221 (0.286) | −0.665 (0.0064*) | – | 0.083 |
| | miR-23a-3p | 0.082 (0.366) | 0.015 (0.865) | – | – | 0.211 (0.257) | −0.052 (0.801) | −0.604 (0.013) | – | 0.082 |
| | miR-24-3p | 0.062 (0.503) | 0.026 (0.784) | – | – | 0.154 (0.421) | 0.125 (0.558) | −0.229 (0.355) | – | 0.027 |

**Notes:**
[1] Males vs. females.
[2] Kazakhs vs. others.
[3] Groups from first column.
[4] Insulin vs. other.
* p-value remains significant after FDR correction for multiple comparisons.

while categorical variables were gender, ethnicity (Kazakh vs. others), and medications taken (insulin vs. other). Other variables were excluded due to their correlation with the selected ones (e.g., age at disease onset), incompleteness (e.g., glycated hemoglobin levels and glomerular filtration rate), or being categorical variables from medical history (e.g., alcohol consumption, smoking, family history of diabetes). All resulting models met the criteria for the absence of autocorrelation and multicollinearity among variables.

Although the resulting models explained only a small proportion of the overall variation (which is typical for such models), in most cases, the primary factor influencing miRNA levels among the tested variables was disease status, i.e., the difference between the studied groups. In some instances, evidence suggested the influence of age and ethnicity; however, in all these cases, this did not significantly affect the importance of the group difference factor.

Thus, disease status was identified as an independent and primary factor among the considered variables influencing the levels of dysregulated miRNAs.

## ROC analysis

To assess the applicability of miRNAs that exhibited differential expression between groups as markers of pathological changes, we conducted ROC analysis. The results of the ROC analysis are presented in Table 6. None of the classifiers examined reached the "Good" rating according to the grading provided by *Muller et al. (2005)*. In most cases, the AUC was in the range of 0.6–0.7, corresponding to "Poor" classifiers. This category included miR-423-3p when distinguishing DR patients from noDM patients (Fig. 2A), as well as three out of four miRNAs when distinguishing DR patients from healthy controls.

**Table 6 ROC analysis results.**

| Classes | Potential markers | AUC (95% CI) | Optimal cut-off point | Sensitivity (95% CI) | Specificity (95% CI) | Accuracy |
|---|---|---|---|---|---|---|
| DR patients *vs.* noDR patients | miR-423-3p | 0.653 [0.576–0.729] | −2.377 | 0.870 [0.788–0.929] | 0.398 [0.300–0.502] | 0.636 |
| NPDR patients *vs.* noDR patients | miR-21-5p | 0.626 [0.525–0.727] | 1.815 | 0.711 [0.541–0.846] | 0.541 [0.437–0.642] | 0.588 |
| | miR-423-3p | 0.633 [0.532–0.735] | −2.651 | 0.711 [0.541–0.846] | 0.571 [0.467–0.671] | 0.610 |
| PDR patients *vs.* PPDR patients | miR-23a-3p | 0.727 [0.587–0.867] | −0.838 | 0.900 [0.683–0.988] | 0.590 [0.421–0.744] | 0.695 |
| PDR patients *vs.* NPDR+PPDR patients | miR-21-5p | 0.644 [0.513–0.775] | 1.650 | 0.591 [0.364–0.793] | 0.744 [0.632–0.836] | 0.710 |
| | miR-23a-3p | 0.705 [0.577–0.832] | −0.838 | 0.900 [0.683–0.988] | 0.532 [0.415–0.647] | 0.608 |
| DR patients *vs.* controls | miR-423-3p | 0.720 [0.622–0.817] | −2.773 | 0.550 [0.447–0.650] | 0.867 [0.693–0.962] | 0.623 |
| | miR-221-3p | 0.694 [0.590–0.798] | 0.100 | 0.464 [0.362–0.568] | 0.893 [0.718–0.977] | 0.560 |
| | miR-23a-3p | 0.642 [0.529–0.755] | −0.608 | 0.701 [0.600–0.790] | 0.571 [0.372–0.755] | 0.672 |
| noDR patients *vs.* controls | miR-24-3p | 0.619 [0.510–0.728] | 0.295 | 0.367 [0.272–0.471] | 0.833 [0.653–0.944] | 0.476 |
| | miR-221-3p | 0.630 [0.520–0.740] | 0.762 | 0.694 [0.593–0.783] | 0.571 [0.372–0.755] | 0.667 |
| | miR-23a-3p | 0.671 [0.566–0.777] | −0.621 | 0.735 [0.636–0.819] | 0.571 [0.372–0.755] | 0.699 |
| T2DM patient *vs.* controls | miR-423-3p | 0.626 [0.536–0.717] | −2.773 | 0.455 [0.384–0.527] | 0.867 [0.693–0.962] | 0.509 |
| | miR-221-3p | 0.662 [0.564–0.759] | 0.125 | 0.405 [0.336–0.478] | 0.893 [0.718–0.977] | 0.466 |
| | miR-23a-3p | 0.657 [0.556–0.758] | −0.608 | 0.718 [0.649–0.780] | 0.571 [0.372–0.755] | 0.700 |

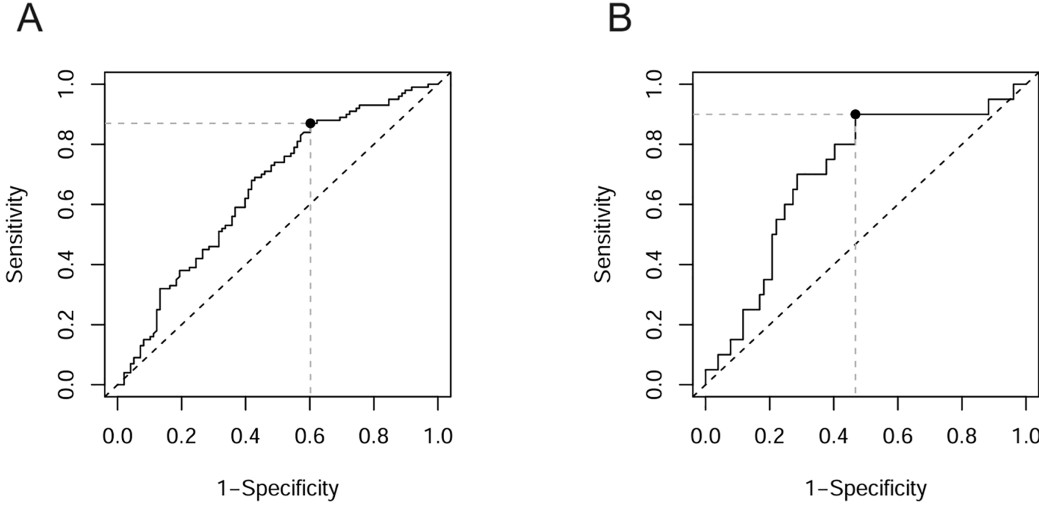

**Figure 2 ROC curves with optimal cut-off points.** (A) miR-423-3p in distinguishing DR patients among T2DM patients; (B) miR-23a-3p in distinguishing PDR patients among DR patients.

Only in three cases did the AUC exceed the threshold of 0.7, corresponding to a "Fair" rating. In two of these cases, the classifier was miR-23a-3p when distinguishing PDR patients from PPDR patients and from PPDR + NPDR patients (Fig. 2B) (AUC = 0.727 and 0.705, respectively). The "Fair" rating was also achieved by miR-423-3p when distinguishing DR patients from healthy controls (AUC = 0.720).

## DISCUSSION

In this study, we examined the levels of plasma miRNAs in patients with T2DM and DR in the population of Kazakhstan. Despite the promising results obtained for the studied miRNAs in earlier works, according to our data, none of them has sufficient potential as a diagnostic marker for DR.

Only miR-423-3p showed a significant difference (decrease) in DR patients compared to diabetes patients without fundus complications; however, the differences were not distinct enough to support its use as a reliable biomarker for the diagnosis of DR. Previously, miR-423-3p has primarily been studied in the context of cancer development, with some studies highlighting its oncogenic role (*Li et al., 2015*; *Zhao et al., 2015*; *Wang et al., 2019*), while others suggest a protective role (*Xue et al., 2022*; *Yan et al., 2022*). Furthermore, miR-423-3p has been identified as a potential plasma biomarker for cancer (*Zhu et al., 2017*; *Guo et al., 2021a*). Several earlier studies have indicated that miR-423-3p may also be involved in the pathogenesis of T2DM and its complications (*Ortega et al., 2014*; *Meerson et al., 2019*; *Blum et al., 2019*). The first two studies demonstrated reduced plasma levels of miR-423-3p in T2DM patients. Our findings align with this, revealing a further decrease in miR-423-3p levels in DR patients. Moreover, we observed a gradual reduction miR-423-3p corresponding to increasing DR severity, which was supported by a weak but consistent correlation. Our findings are in agreement with those of *Blum et al. (2019)*, who reported a negative correlation between miR-423-3p levels and DR progression in an Israeli population. Conversely, another study conducted in the same population found a nonlinear association for miR-423-3p, with levels decreasing in the early stages of T2DM and increasing in later stages with complications (*Meerson et al., 2019*). *Blum et al. (2019)* further proposed that miR-423 may contribute to the regulation of diabetic retinal vascular proliferation through its interactions with vascular endothelial growth factor (VEGF) and endothelial nitric oxide synthase (eNOS).

Consistent with previous studies, our results suggest that miR-23a-3p, miR-221-3p, and miR-21-5p may also be involved in the development and progression of T2DM.

According to the literature, miR-221-3p plays a significant role in angiogenesis and may therefore be altered in diabetes complications (*Liu et al., 2018*; *Wang et al., 2020a*; *Yu et al., 2020b*; *Yu et al., 2022*; *Zhao & Pan, 2023*; *Gentile et al., 2023*; *Hu et al., 2024*). It seems logical that in diabetic foot ulcers elevated levels of miR-221-3p contribute to wound healing (through its effects on homeodomain-interacting protein kinase 2 (HIPK2) (*Yu et al., 2022*) and the AKT/eNOS (*Yu et al., 2020b*) and DYRK1A/STAT3 signaling pathways (*Hu et al., 2024*), whereas in DR, conversely, miR-221-3p promotes microvascular dysfunction (by affecting tissue inhibitor of metalloproteinase 3 (TIMP3) (*Wang et al., 2020a*)), with its serum level positively correlating with VEGF levels and disease progression (*Liu et al., 2018*; *Zhao & Pan, 2023*). In our study, we found a weak negative correlation between the level of miR-221-3p and the progression of T2DM, although significant differences in means between groups were observed only regarding the presence/absence of diabetes, not regarding the presence/absence of retinopathy. Additionally, a weak negative correlation with age of diabetes onset was observed. Our data

are more consistent with studies of T2DM in relation to diabetic foot ulceration, where patients are expected to have decreased expression of miR-221-3p, resulting in non-healing wounds.

The level of circulating miR-23a-3p was also significantly reduced in T2DM patients in the Kazakhstani population. Our results are quite consistent with previous studies. In accordance with the literature, T2DM patients exhibit reduced levels of miR-23a-3p in adipose tissue (*Lozano-Bartolomé et al., 2018*), blood (*Chang et al., 2021*) and blood serum (*Yang et al., 2014*), leading to tumor necrosis factor (TNFα)-induced insulin resistance (*Lozano-Bartolomé et al., 2018*) and overexpression of its target NIMA-related kinase 7 (NEK7) in macrophages, and consequently, NLRP3-induced inflammation (*Chang et al., 2021*). Further reduction in plasma miR-23a-3p levels is observed in diabetic kidney disease (*Meng, Yu & Lei, 2023*) and DR at both early (*Mastropasqua et al., 2021*) and late stages (*Sun, Liu & Zuo, 2021*). As shown by *Sun, Liu & Zuo (2021)*, downregulation of miR-23a-3p in the retina of diabetic patients can promote the development of retinopathy through the overexpression of its target VEGF. In agreement with this, we identified a significant reduction of miR-23a-3p levels in PDR patients. However, we did not observe such a reduction in patients with NPDR and PPDR, which is inconsistent with the findings of *Mastropasqua et al. (2021)*. It is also necessary to mention the work of *Santovito et al. (2021)*, the only study that reported an increase (not a decrease) in plasma miR-23a-3p levels in DR patients.

miR-21-5p is perhaps the most frequently dysregulated miRNA in various diseases (*Jenike & Halushka, 2021*), including T2DM and complications (*Rezazadeh-Gavgani et al., 2023*; *Qing et al., 2014*; *Chen et al., 2017*; *Jiang et al., 2017*; *Lou et al., 2019*; *Grieco et al., 2022*). Accumulated knowledge suggests the involvement of miR-21-5p in inflammatory processes through the regulation of nuclear factor kappa B (NF-κB) and nucleotide-binding oligomerization like receptor family pyrin domain containing 3 (NLRP3) pathways (*Olivieri et al., 2021*), explaining its broad spectrum of diseases in which miR-21-5p dysregulation may occur. Previous studies have shown that miR-21-5p plays a role in the pathogenesis of T2DM by influencing Nuclear Factor Kappa B Subunit 1 (NFKB1) and TP53-inducible glycolysis and apoptosis regulator (TIGAR) targets in visceral adipose tissue (*Wróblewski et al., 2023*), and in the development of retinopathy in T2DM by affecting Peroxisome proliferator-activated receptor alpha (PPAR-α) and Growth factor-beta (TGF-β) signaling pathways (*Chen et al., 2017*; *Lou et al., 2019*). In our study, we found a significant increase in plasma levels of miR-21-5p in patients at the early stages of DR (NPDR), but not at later stages (PDR). The observed positive correlation with disease duration and a negative correlation with age of diabetes onset in noDR patients indicate an association of plasma miR-21-5p levels with the development of T2DM. However, according to our data, miR-21-5p cannot be used as a biomarker for T2DM and complications, despite previous suggestions of its potential in this role (*Qing et al., 2014*; *Jiang et al., 2017*; *Grieco et al., 2022*). Another obstacle to using of miR-21-5p as a biomarker is its lack of specificity to a particular disease; rather, it should be considered as a general marker of inflammation (*Jenike & Halushka, 2021*).

Unfortunately, despite the identified patterns of changes in circulating miRNAs in T2DM and DR, the conducted ROC analysis did not reveal good candidates for biomarkers of the studied pathological conditions. According to our data, only circulating miR-23a-3p can be considered as a potential minimally invasive biomarker with moderate informativeness for diagnosing PDR (although this result needs to be verified on a larger sample of PDR patients, as evidenced by the wide confidence interval for AUC), while miR-423-3p, miR-221-3p, miR-23a-3p, and miR-21-5p are unlikely to be useful in the diagnosis of T2DM or DR. Additionally, we consider it important to note the small magnitude of the observed differences in the levels of the studied miRNAs between groups (differences did not exceed 1.5-fold values). The small magnitude of changes between groups obviously complicates classification, even in the case of fairly specific differences. In summary, we conclude that circulating plasma miRNA molecules have low potential as biomarkers for DR.

Previous studies have shown that interpopulation differences may be one of the reasons for the poor reproducibility of such research (*Fickweiler et al., 2023*). Our multiple regression analysis revealed that age and ethnicity could influence plasma miRNA levels. Standardizing the study protocol can address the issue of inconsistent experimental conditions; however, biological factors must be carefully considered, and previously recommended biomarkers should be validated within specific populations.

## CONCLUSIONS

In our study on the Kazakhstani population, we identified dysregulation of several miRNAs in plasma of T2DM patients with and without DR. Our data generally support the conclusions of earlier works regarding the significant role of the miRNAs in the pathogenesis of diabetes and its complications. However, the low specificity and small magnitude of its level deviations do not allow us to recommend these miRNAs as biomarkers for diagnosing of DR.

### Funding

This research was funded by the Committee of Science of the Ministry of Science and Higher Education of the Republic of Kazakhstan (Grant No. BR27195585). The funders had no role in study design, data collection and analysis, decision to publish, or preparation of the manuscript.

### Grant Disclosures

The following grant information was disclosed by the authors:
Committee of Science of the Ministry of Science.
Higher Education of the Republic of Kazakhstan: BR27195585.

### Competing Interests

The authors declare that they have no competing interests.

## Author Contributions

- Aizhan Magazova conceived and designed the experiments, performed the experiments, analyzed the data, prepared figures and/or tables, authored or reviewed drafts of the article, and approved the final draft.
- Yeldar Ashirbekov conceived and designed the experiments, analyzed the data, prepared figures and/or tables, authored or reviewed drafts of the article, and approved the final draft.
- Arman Abaildayev conceived and designed the experiments, performed the experiments, prepared figures and/or tables, authored or reviewed drafts of the article, and approved the final draft.
- Kantemir Satken performed the experiments, prepared figures and/or tables, authored or reviewed drafts of the article, and approved the final draft.
- Gulzhakhan Utegenova performed the experiments, authored or reviewed drafts of the article, and approved the final draft.
- Ayaz Belkozhayev analyzed the data, authored or reviewed drafts of the article, and approved the final draft.
- Altynay Balmukhanova analyzed the data, authored or reviewed drafts of the article, and approved the final draft.
- Zaure Dzhumatayeva performed the experiments, authored or reviewed drafts of the article, and approved the final draft.
- Ainagul Beissova performed the experiments, authored or reviewed drafts of the article, and approved the final draft.
- Iryna Shargorodska conceived and designed the experiments, authored or reviewed drafts of the article, and approved the final draft.
- Aigul Balmukhanova conceived and designed the experiments, authored or reviewed drafts of the article, and approved the final draft.
- Kamalidin Sharipov conceived and designed the experiments, authored or reviewed drafts of the article, project administration, supervision, and approved the final draft.

## Human Ethics

The following information was supplied relating to ethical approvals (*i.e.*, approving body and any reference numbers):

The study was approved by the local ethics committee of Aitkhozhin Institute of Molecular Biology and Biochemistry (No. 4.60/01-03)

## Data Availability

Clinicopathological data of patients and experimental data are available in the Supplemental File 1. The results from the NormFinder and geNorm programs (for determining the stability of normalizers) are available in the Supplemental File 2.

## Supplemental Information

Supplemental information for this article can be found online at http://dx.doi.org/10.7717/peerj.19259#supplemental-information.

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
