# Peer review of "Circulating microRNAs demonstrate limited diagnostic potential for diabetic retinopathy in the population of Kazakhstan"

_PeerJ, doi:10.7717/peerj.19259_

## Round 0.1 · original submission · Major Revisions

Please revise your manuscript according to the reviewers' comments.
Yours,
Yoshi
Prof. Yoshinori Marunaka, M.D., Ph.D.

·

Basic reporting

In The manuscript, the authors did interesting work and manuscript is well written and solid results

Experimental design

Good designed

Validity of the findings

In The manuscript, the authors showed solid results

Additional comments

Thanks for inviting me to review this valuable manuscript. This article is going to evaluate the Circulating microRNAs as potential biomarkers for diabetic retinopathy in the population of Kazakhstan.

I sincerely think that the authors took care in describing the experimental methods. The results and the forthcoming conclusions are well structured and presented.
In consequence, the article has enough quality to be considered for publication.

Just as minor points to be considered:

1- In results, authors reported that microRNA Reduced in comparing between groups please add more details about how these marker could affect development of DM or DR.
2- Please Add more details about method of calculation of RNA relative expression The comparative ΔΔCt method

·

Basic reporting

no comment

Experimental design

The study is designed well, but there are some points in methodology:

The important point in the demography of diabetic patients is that the average duration of T2DM onset is about 12 years, but the only characteristic considered for their diabetes is a blood sugar level, which is normal in patients. This is a fundamental question raising doubt about the main association findings of miRNA and retinopathy. why is HbA1c not considered for approval for diabetes.? Why is there no history of diabetic medications except insulin therapy as it is T2DM?

- The collecting sample comes years, and there are several new studies already approving this associtionship, although we know it is related to the population of Khazakh. Is there any rationale for this delay? is there any comment in the discussion to explain the differences in finding with other studies in the Kazakh population?

Validity of the findings

although the conclusion mentioned the lack of accurate association between miRNA and retinopathy, the low sample of this disease can be a limiting factor. it should mentioned and discussed because diabetes is not a low-frequency disorder, and much higher samples in similar studies found an association.

Reviewer 3 ·

Basic reporting

"Circulating microRNAs as potential biomarkers for diabetic
retinopathy in the population of Kazakhstan"
Very confusing title because the study concludes that the selected microRNAs, particularly miR-23a-3p, exhibit low specificity and a small magnitude of changes when comparing levels between different patient groups. This diminishes their potential effectiveness as reliable biomarkers for diagnosing DR.

Experimental design

Line 45-46: not clear
Line 127: were collected between 2020-2021: It was in the peak of the COVID outbreak, did they check for infection before collecting the samples
Line 159: missing catalog number
Line 162: missing catalog number
Line 165: missing catalog number
Line 171: ΔCt values (ΔCt = Ctreference − Cttarget). Is it correct or something is missing please check the formula https://pmc.ncbi.nlm.nih.gov/articles/PMC4280562/
Line 171: Endogenous miR-191-5p was used as a reference. Why? https://www.biovendor.com/mir-191-5p
researchers often use small nuclear RNAs (snRNAs) as reference genes.
Line 174: was considered as log2 fold change. Why is 2-ΔΔCT not used?
Line 178: Student's t-test for quantitative data. I recently made the mistake of using a t-test for comparing multiple groups and found out it’s not correct, please check, 1 or 2-way ANOVA is a better method.
Line 208: "qRT-PCR"
Line 210:one exogenous control: is it used in the calculation?
Line 398: Believe is a strong word for a scientific paper; it just needs technological advancement. Passing verdicts might not be a good statement.
Table 2: p-value and FDR are different, in places FDR is used but in the table it's the p-value, please check
Figure1: Y axis is missing

Validity of the findings

The problem of variability comes from various factors age, diet, duration of DR, ethnicity, etc, but the paper lacks explaining these factors.
The authors identified some differences in miRNA levels, but none retained significance after adjustments for multiple comparisons. This indicates that the findings may not be robust. Please check qRT-PCR normalization and calculation methods.

---

## Round 0.2 · accepted · Accept

Congratulations!

Yours,

Yoshi

Prof. Yoshinori Marunaka, M.D., Ph.D.

·

Basic reporting

No comment

Experimental design

No comment

Validity of the findings

No comment